# Diverging Effects of Adolescent Ethanol Exposure on Tripartite Synaptic Development across Prefrontal Cortex Subregions

**DOI:** 10.3390/cells11193111

**Published:** 2022-10-02

**Authors:** Christopher Douglas Walker, Hannah Gray Sexton, Jentre Hyde, Brittani Greene, Mary-Louise Risher

**Affiliations:** 1Department of Biomedical Research, Joan C. Edwards School of Medicine, Marshall University, Huntington, WV 25701, USA; 2Neurobiology Research Laboratory, Hershel ‘Woody’ Williams Veterans Affairs Medical Center, Huntington, WV 25704, USA

**Keywords:** astrocyte, adolescent, ethanol, prefrontal cortex, tripartite synapse

## Abstract

Adolescence is a developmental period that encompasses, but is not limited to, puberty and continues into early adulthood. During this period, maturation and refinement are observed across brain regions such as the prefrontal cortex (PFC), which is critical for cognitive function. Adolescence is also a time when excessive alcohol consumption in the form of binge drinking peaks, increasing the risk of long-term cognitive deficits and the risk of developing an alcohol use disorder later in life. Animal models have revealed that adolescent ethanol (EtOH) exposure results in protracted disruption of neuronal function and performance on PFC-dependent tasks that require higher-order decision-making. However, the role of astrocytes in EtOH-induced disruption of prefrontal cortex-dependent function has yet to be elucidated. Astrocytes have complex morphologies with an extensive network of peripheral astrocyte processes (PAPs) that ensheathe pre- and postsynaptic terminals to form the ‘tripartite synapse.’ At the tripartite synapse, astrocytes play several critical roles, including synaptic maintenance, dendritic spine maturation, and neurotransmitter clearance through proximity-dependent interactions. Here, we investigate the effects of adolescent binge EtOH exposure on astrocyte morphology, PAP-synaptic proximity, synaptic stabilization proteins, and dendritic spine morphology in subregions of the PFC that are important in the emergence of higher cognitive function. We found that adolescent binge EtOH exposure resulted in subregion specific changes in astrocyte morphology and astrocyte-neuronal interactions. While this did not correspond to a loss of astrocytes, synapses, or dendritic spines, there was a corresponding region-specific and EtOH-dependent shift in dendritic spine phenotype. Lastly, we found that changes in astrocyte-neuronal interactions were not a consequence of changes in the expression of key synaptic structural proteins neurexin, neuroligin 1, or neuroligin 3. These data demonstrate that adolescent EtOH exposure results in enduring effects on neuron-glia interactions that persist into adulthood in a subregion-specific PFC manner, suggesting selective vulnerability. Further work is necessary to understand the functional and behavioral implications.

## 1. Introduction

Despite declining rates of alcohol consumption since 1975, alcohol remains the most commonly used licit substance and a leading cause of death and injury in the United States [1], while alcohol-related deaths have been steadily rising by ~2.2% per year over the past two decades [2]. According to the World Health Organization, 3 million deaths every year result from the harmful use of alcohol, representing 5.3% of all deaths worldwide [3]. The emergence of the COVID-19 pandemic has compounded this problem resulting in an increase of 25.5% in alcohol-related deaths during the period 2019–2020 [4].

In the United States, alcohol consumption typically begins and escalates during adolescence. Although adolescents drink less frequently than adults, adolescents typically consume more alcohol per occasion, most commonly in a binge-like manner [5,6]. Early alcohol use, in the form of binge drinking, increases the risk of acute and chronic physical and mental health problems, as well as alcohol dependence later in life [7,8,9,10]. It has been proposed that this is, in part, due to the ongoing development of the adolescent brain, in which excessive alcohol use drives maladaptive neuronal changes that alter brain connectivity and result in remodeling of the reward network (for a review, see Koob and Volkow [11]).

Select cortical regions, such as the prefrontal cortex (PFC) are known to undergo the final stages of neuronal maturation during adolescence, occurring via the pruning of synapses and the refinement of neuronal circuitry. Through this process, local and projecting neuronal networks are refined and stabilized. The PFC is composed of multiple subregions, including the anterior cingulate cortex (ACC), the medial prefrontal cortex (mPFC), and the orbitofrontal cortex (OFC) that play distinct roles in higher-order executive functions such as reasoning, planning, language, and social interactions [12], all of which are important aspects of the addiction/reward circuitry. The ACC has been associated with complex cognitive functions such as empathy, impulse control, emotion, and decision-making [13], while impaired ACC function is associated with psychopathology and emotional dysregulation [14]. The mPFC is essential in regulating cognitive function, including attention, inhibitory control, habit formation, and working, spatial, and long-term memory [15]. The mPFC is also important in the regulation of conditioned behavior and suppression of these behaviors [16], with the mPFC’s prelimbic cortex (PL) and infralimbic cortex (IL) being strongly associated with the activation and suppression of fear circuits [17,18]. The OFC is associated with sensory integration and modulation of visceral reactions, participation in learning, and decision-making for emotional, adaptive, and goal-directed behavior [19]. Within the OFC the lateral (sometimes referred to as the ventrolateral (LO-OFC)) and ventral (VO-OFC) aspects appear to have diverging roles. The LO-OFC is involved in obsessive compulsive behavior [20] and decision making through the integration of immediate prior information and current information [21], while the VO-OFC appears to be involved in value assessment in tasks such as delayed discounting [15]. Damage or degeneration of the PFC can manifest as a deficit in impulse control, poor performance on tasks that require long-term planning, blunted emotional responses, aggression, and irritability [22]. Therefore, it is not surprising that PFC dysfunction has been implicated in many human psychiatric disorders, the emergence and persistence of drug and alcohol use, and as a predictor of increased risk to drug and alcohol relapse [23].

Previous studies have shown a positive correlation between ACC activation and excessive drinking [24], while the mPFC has been implicated in the development and persistence of addictive behaviors through reinforcement of rewarding stimuli, compulsive drug taking, drug-associated memories, and relapse [25,26,27]. The OFC, which is critical for impulse control, has shown to be impaired in individuals with alcohol use disorders [28,29] however the contributions of OFC subregions have not been explored. Taken together, these data provide a rationale for continued investigation into the impact of adolescent binge alcohol exposure on PFC development, contribution to the emergence of long-term PFC-dependent neurological deficits [30], and increased propensity to develop alcohol dependence [31].

Despite strong evidence of neuronal deficits following adolescent binge alcohol exposure across many PFC subregions, much of this research ignores the potential contribution of astrocytes to these cognitive impairments. Growing evidence is beginning to reveal the critically important role of astrocytes across numerous domains of cognition and a wide variety of brain regions including PFC subregions (for a review, see Lyon & Allen [32]), as such the involvement of the ACC in remote memory tasks and novel open field exploratory behavior [33,34]. Interestingly, PFC astrocytes have also been implicated in bidirectionally modulating ethanol (EtOH) consumption in male mice [35].

Astrocytes are complex non-neuronal glial cells with extensive peripheral astrocytic processes (PAPs) that ensheathe presynaptic axonal and postsynaptic dendrite components to form the ’tripartite synapse’ (for a review, see Walker et al. [36]). It is estimated that a single astrocyte can be connected with up to 2 million synapses in humans [37]. This connectivity establishes a wide-ranging network, allowing a single astrocyte to integrate and influence neuronal activity across independent circuits [37]. The role of astrocytes at the tripartite synapse is multifaceted (for reviews, see Farhy-Tselnicker & Allen, 2018; Kim et al., 2017; Lyon & Allen, 2021 [32,38,39]). However, many astrocyte functions involve contact mediated signaling or the release of astrocyte secreted signaling factors, both of which rely heavily on appropriate PAP-synaptic proximity and localization. For example, astrocytes are heavily involved in dendritic spine maturation through direct contact mediated signaling [40] and the release of astrocyte secreted factors [41]. Moreover, the further away the PAP is from the synapse, the more likely the dendritic spine will be structurally immature [42]. These data suggest that tight regulatory control of PAP-synaptic proximity may be critical for appropriate dendritic spine maturity.

PAP-synaptic proximity relies on the availability of stabilizing proteins. For example, disruption of eph-ephrin located on astrocyte processes and dendritic spines, respectively, has been shown to be important for dendritic stabilization [40]. More recently, the role of the cell adhesion molecules, neuroligins (located on postsynaptic terminals and astrocytes depending on the subtype) and neurexins (located on presynaptic terminals) have been shown to play a critical role in astrocyte complexity, synaptic contact, and synaptic function [43]. Moreover, disruption of neurexin binding prevented increases in pre- and post-synaptic size [44], suggesting that the ability of neurexin to bind its appropriate synaptic partner is critical for dendritic spine and overall synaptic maturity.

It is well established that neuronal maturation is ongoing within the PFC throughout adolescence and into early adulthood. Recently, we demonstrated that the relationship between synapses and astrocytes also continues to develop throughout adolescence into early adulthood in the mPFC (IL and PL), as demonstrated by an increase in PAP-synaptic interactions/proximity across this time period [45]. Given the emerging appreciation for the critical role that astrocytes play in synaptic maturation, regulation, and cognition, and the critical PAP-synaptic developmental processes that are ongoing during adolescence, we sought to investigate the impact of adolescent binge ethanol exposure on astrocyte morphology, PAP-synaptic proximity, dendritic spine maturation, and determine whether loss of neuroligin-neurexin cell adhesion proteins was a driving factor in the observed changes. We hypothesize that adolescent intermittent binge EtOH exposure (AIE) disrupts PAP-synaptic proximity in a region-dependent manner and is associated with changes in dendritic spine maturation.

## 2. Methods

All procedures used in this study were conducted in accordance with the guidelines of the American Association for the Accreditation of Laboratory Animal Care and the National Research Council’s Guide for Care and Use of Laboratory Animals and were approved by the Huntington VA Medical Center and Marshall University.

### 2.1. Animals and Surgical Procedures

A total of 46 male Sprague Dawley rats (Hilltop, Scottdale, PA, USA) were received on postnatal day (PND) 21 and were double-housed and maintained in a temperature- and humidity-controlled room with access to food and water ad libitum. All animals, regardless of treatment group or experiment, were housed together in a randomly assigned order on a 12 h:12 h reverse light:dark cycle (lights on at 6:00 p.m.) and allowed to acclimatize for 5 days. Surgical procedures were performed as previously described in Testen et al., 2019 [45] with minor modifications. On PND 26, rats (n = 10/treatment group, total = 40 rats) were administered mannitol (0.3 mL/kg, 25% *w*/*v*, i.p.) [45,46] and anaesthetized with a cocktail of ketamine (30 mg/ kg), xylazine (2.5 mg/kg), and acepromazine (0.5 mg/kg) before being secured to the stereotax (Figure 1A). For visualization of astrocytes, a green fluorescent protein (GFP) with the lymphocyte protein tyrosine kinase (Lck) tag was used to enhance the visualization of the finer PAPs [47,48,49]. Lck-GFP was expressed under the control of the astrocyte-specific GfaABCID promotor using adeno-associated virus (AAV) type 5 (a gift from Katie Reissner (UNC Chapel Hill) and purchased from Addgene Plasmid #105598, RRID:Addgene_105598) [45,48,50,51]. The AAV was microinjected with a 26-gauge needle (1.0 μL per side, 7.3 × 10^12^ particles/mL at 1 μL /min) with a dwell time of 10 min at the following coordinates: +2.7 mm anterior/posterior, +/−0.6 mm medial/lateral, −3.8 mm dorsal/ventral for mPFC and ACC and −5.2 mm anterior/posterior, +/−2.5 mm medial/lateral, −2.9 mm dorsal/ventral for OFC (Figure 1B). No animals died during the surgery/post-surgical period. Following surgery, animals were double-housed with their original cagemate and randomly assigned into experimental groups (by cage) using an online random number generator, in preparation for binge EtOH exposure. Experimenters were blinded for all experiments and data collection. Analysis was conducted with letter assignments ensuring that treatment groups remained unknown. All animal numbers are described for the individual experimental groups below and based on a prior power analysis using G*Power version 3.1.9.4 (Axel Buchner, Heinrich Heine University, Dusseldorf, Germany) for sample size estimation. This analysis was based on data from previously published and pilot data with a significance criterion of α = 0.05 and power β = 0.8 to determine the required minimum sample size.

### 2.2. Intermittent Binge EtOH Exposure

Prior to EtOH/water (H_2_O) exposure, animals were habituated to handling. Beginning PND 30, animals were exposed to EtOH or H_2_O consisting of 10 doses of 5 g/kg EtOH (35% *v*/*v* in H_2_O) or H_2_O by intragastric gavage (i.g.) using 2 days on, 1 day off, 2 days on, 2 days off intermittent schedule of 16 days (Figure 1A), as previously described in Risher et al., 2015 [52]. While all experimenters were blinded, this is difficult to maintain during EtOH administration due to the strong smell of EtOH. The experimenter that administers the EtOH is also responsible for ensuring that the animals recover from intoxication, once again making blinding at this stage of the experiment difficult. The experimenter conducting the EtOH administration was responsible for ensuring that everyone involved in the experiments listed below remained blinded until completion of the study. Animals were euthanized 24 h after the 10th/last dose (PND 46) or following a 26-day forced abstinence period (PND 72), allowing animals to reach adulthood prior to tissue collection. EtOH doses were selected to produce blood EtOH concentrations (BECs) consistent with those of adolescent humans that are achieved during binge drinking episodes [53].

### 2.3. Immunohistochemistry (IHC)

#### 2.3.1. Slice Preparation

At PND 45 or 72, animals were anesthetized with isoflurane (10 animals/treatment group, total = 40), then perfused with phosphate-buffered saline (PBS, pH7.4; Sigma P5368) for 5 min followed by 4% paraformaldehyde (PFA; cat# 19210, Electron Microscopy Solutions, Hatfield, PA, USA) for 12 min (20 mL/min). Brains were harvested and post-fixed in 4% PFA at 4 °C overnight followed by 30% glycerol (cat# G5516-1L, Sigma-Aldrich, St Louis, MO, USA) in PBS at 4 °C for 1–2 days. Brains were placed in 2:1 of 30% sucrose to OCT freezing compound (Electron Microscopy Sciences, Hatfield, PA, USA) and stored at −80 °C. 80 μm slices were collected using a cryostat (CM 1950, Leica Biosystems, Richmond, IL, USA). Sections underwent antigen retrieval as described in [45,54]. Briefly, free-floating sections were rinsed three times (5 min each) in 0.1 M PB (3.1 g/L NaH_2_PO_4_, 10.9 g/L Na_2_HPO_4_, pH 7.4) then transferred to 10mM sodium citrate buffer preheated to 80 °C for 30 min, shaking the slices every 10 min. Sections were cooled and washed 3 times (5 min each) in 0.1M PB (pH 7.4).

#### 2.3.2. PSD-95 and GFAP

Sections were blocked with 5% normal goat serum (NGS, Jackson Immunolabs,, West Grove, PA, USA, cat# 005-000-121) in PBST (2% triton 100-X, Roche, cat#13134900) for one hour at room temperature. Sections were then incubated for four days in mouse anti-PSD-95 (1:450 Thermo Fisher Scientific Cat# MA1-045, RRID:AB_325399) and rabbit anti-GFAP (1:500 Agilent Cat# Z0334, RRID:AB_10013382) in PBST (2% triton, 5% NGS) at 4 °C. Sections were washed three times in PBST (0.2% triton) and incubated for 6 h at room temperature in Alexa Fluor goat anti-mouse 594 (1:200 Molecular Probes Cat# A-11032, RRID:AB_2534091) and Alexa Fluor goat anti-rabbit 647 (1:200 Thermo Fisher Scientific Cat# A-21245, RRID:AB_2535813) in PBST (2% triton X-100, 5% NGS). Sections were washed three times in PBST (0.2% triton) and once in PBS. Slices were mounted with Vectashield+DAPI (Vector Laboratories West Grove, PA, USA, cat# H-1200) and coverslips were sealed with nail varnish.

#### 2.3.3. Neuroligin 1, 3, and Neurexin

Sections were blocked with 5% normal natural donkey serum (NDS, Jackson Immunolabs, cat# 005-000-121) in PBST (2% triton 100-X, Roche, cat# 13134900) for one hour at room temperature. Then, incubated for four days in rabbit anti-Neurexin ½/3 (1:500 Synaptic Systems Cat# 175 003, RRID:AB_10697815) with either mouse anti-Neuroligin 1 (1:500 Synaptic Systems, cat# 129111) or mouse anti-Neuroligin 3 (1:500 Synaptic Systems Cat# 129 111, RRID:AB_887747) in PBST (2% triton, 5% NDS) at 4 °C. Sections were washed three times in PBST (0.2% triton) and incubated for 6 h at room temperature in Alexa Fluor goat anti-mouse 594 (1:200) and Alexa Fluor goat anti-rabbit 647 (1:200) in PBST (2% triton X-100, 5% NDS). Sections were washed three times in PBST (0.2% triton) and once in PBS. Slices were mounted with Vectashield+DAPI and coverslips were sealed with nail varnish.

### 2.4. Data Acquisition and Processing

#### 2.4.1. Astrocyte-Synaptic Co-Localization

A Leica SP5 laser-scanning confocal microscope with 63× oil-immersive objective, NA 1.45 (Leica, Wetzlar, Germany) was used for image acquisition. Acquisition parameters for AAV^+^ astrocyte-PSD-95 imaging were set at 1024 × 1024 pixels frame size, 16-bit depths, 4× lines averaging, 1 μm z-step size. Lck-GFP expression pattern is diffuse, aiding in the acquisition of single, isolated astrocytes. Individual, whole GFP^+^ astrocytes were randomly selected by the experimenter. GFP^+^, GFAP, and PSD-95 were then imaged within the region of interest: ACC, mPFC (infralimbic (IL) and prelimbic (PL)), and OFC (ventral orbital area (VO-OFC) and lateral orbital area (LO-OFC)). 8–11 individual astrocytes were captured per brain region. A total of 5 brains were used per treatment group.

Co-localization was performed as previously described in Testen et al., 2019 [45], with modifications. AutoQuant X3.1.2 software (Media Cypernetics, Rockville, MD, USA) was used to deconvolve raw images before digitally reconstructing z-stacks (Figure 1C). AutoQuant’s algorithm for blind deconvolution with 10 iterations was run on each z-stack prior to reconstruction. Parameters for blind deconvolution are automatically optimized by the software based on confocal, objective, and imaging specifications. Output files were directly imported to Imaris x64 (Bitplane, Santa Barbara, CA, USA) for 3-dimensional reconstruction. Using Imaris, each individual astrocyte was first isolated from a Lck-GFP background signal using a surface building feature. Surface rendering enables the extraction of morphometric values, including surface area and volume, and generates a new Lck-GFP channel devoid of background noise, revealing only a signal from a single isolated astrocyte (Figure 1C). Close attention was paid to verify that collected Lck-GFP signal accounted for a single astrocyte, in its entirety. Co-localization of the Lck-GFP and GFAP signals was used to confirm astrocyte identity for quantification. Only confirmed astrocytes that were captured in their entirety were used for quantification and analysis. The isolated Lck-GFP channel (surface mask) was used, in conjunction with the PSD-95 channel, to perform co-localization analysis to determine the proximity between the astrocytes and post-synaptic neuronal terminals (Figure 1C). Before co-localization analysis, the threshold for the PSD-95 signal was manually determined by measuring the fluorescence intensity of unambiguous PSD-95 positive puncta on multiple optical planes by a blinded experimenter. This was achieved by rotating the reconstructed astrocyte image in 3D space while adjusting voxels to ensure that PSD-95 expression could be observed without the interference of background noise. An average of these measurements was used as a final PSD-95 signal threshold value. Co-localization is reported as a % of astrocyte volume (identified as Lck-GFP^+^ surface reconstruction) co-localized with the PSD-95 channel.

#### 2.4.2. Neuroligin 1, 3, and Neurexin

Confocal z-stacks (5 µm thick, optical section depth 0.33 µm, and 1024 × 1024 image size) of the ACC, mPFC (IL and PL), LO-OFC, and VO-OFC were imaged on a Leica SP5 laser-scanning confocal microscope with 63× oil-immersive objective, NA 1.45 (Leica, Wetzlar, Germany). A total of 3 randomly selected image stacks from a randomly selected hemisphere were captured from 3 separate brain slices per animal (9–10 brains/treatment group). AutoQuant X3.1.2 software was used to deconvolve raw images before quantifying protein expression and co-localization. Output files were directly imported to Imaris for analysis. Before analysis, the threshold for protein expression was manually determined by measuring fluorescence intensity of unambiguous neuroligin 1, 3, and neurexin markers on multiple optical planes by a blind experimenter. This was achieved by rotating the reconstructed astrocyte image in 3D space while adjusting voxels to optimize the signal for each protein of interest neuroligin 1, 3, and neurexin). An average measurement for each protein of interest was used as the final signal threshold value. The Imaris ‘Spot’ function was used to create individual markers for neuroligin 1, 3, and neurexin. Expression of individual proteins were observed based on the number of individual markers. Neurexin-neuroligin interactions were analyzed based on co-localization of neurexin-neuroligin 1 and neurexin-neuroligin 3. Representative images for publication were created by importing LIF files into ImageJ and selecting the Z-stack of interest. After converting the Z-stack to a maximum projection image, brightness and contrast were adjusted, then the images were saved as TIF files. Finally, the TIF files were uploaded into Photoshop (Adobe, San Jose, CA, USA) and cropped into a 4 × 4 grid section.

#### 2.4.3. Golgi-Cox Staining

Golgi-Cox staining was performed as previously described [52,55]. The animals (PND 72, n = 3/treatment group, total = 6) were deeply anesthetized with isoflurane, decapitated, and the brain was quickly removed. One hemisphere was randomly selected, quickly rinsed in distilled water, and immersed in a 1:1 mixture of solutions A and B (Rapid Golgi Stain Kit; cat#PK401; FD Neurotechnologies, Baltimore, MD, USA). After 2 weeks of impregnation in solutions A and B, brains were transferred to solution C for 48 h, then removed and frozen in tissue freezing medium (cat#72592 Electron Microscopy Sciences, Hatfield, PA, USA). Coronal slices (100 µm) were sectioned using a cryostat and mounted onto 2% gelatin-coated slides (cat#PO101; FD Neurotechnologies, Baltimore, MD, USA). Sections were stained with a mixture containing Rapid Golgi Stain solutions D and E, then dehydrated, cleared, and coverslipped with Permount Mounting Medium (Electron Microscopy Sciences; cat#17986-05).

#### 2.4.4. Dendritic Spine Analysis

Golgi-impregnated neurons were visualized using the Leica Microscope DM5500B, and image stacks were generated using a 63× oil immersion lens. Image stacks were imported into RECONSTRUCT software (available from http://synapses.clm.utexas.edu/tools/index.stm (accessed on 21 March 2022); Fiala, 2005) for analysis as described in Risher and colleagues [52,55] with modifications. Secondary dendritic branches of ACC, mPFC (IL and PL), LO-OFC and VO-OFC neurons were analyzed (blinded) using an unbiased rating system by measuring the length and width of each protrusion with visible connections to the dendritic shaft from dendritic segments 10 µm in length. Average spine densities were calculated using 2 to 3 separate dendrites from at least 4 to 5 separate image stacks per animal (a total of 15 dendritic branches/animal). Spine types were determined on the basis of the ratio of the width (W) of the spine head to the length (L) of the spine neck and classified as (in µm): filopodia (L > 1.5), thin/long thin (L < 1.5 & L:W > 1), stubby (L:W < 1), and mushroom (W > 0.6), also see [55]. Representative images for publication were created by importing TIF file with dendrite of interest into ImageJ. The background was removed, using the ‘Subtract Background’ function with a rolling ball radius of 10 pixels with light background. TIF files were imported into Photoshop where dendrite sections of interest were isolated. Final representative images were created by adjusting brightness and contrast.

#### 2.4.5. Blood EtOH Concentrations (BECs)

To minimize confounds associated with stress and avoid stress/EtOH interaction in our experimental group, a parallel group of animals (n = 6) were dosed separately to assess BECs obtained in our intermittent binge EtOH paradigm. Animals received intermittent oral gavage (as described above) of 5 g/kg EtOH beginning on PND 30. Blood was collected from the lateral saphenous vein 60 min after EtOH administration on the first and last days of the EtOH administration. Blood samples were centrifuged for 5 min then serum was removed and stored at −80C. Samples were analyzed in triplicate using an Analox AM1 alcohol analyzer (Analox Instruments LTD, Stourbridge, UK).

### 2.5. Statistical Analysis

Data were grouped using Excel 365 (Microsoft, Redmond, WA, USA) and analyzed using GraphPad Prism Version 9.3.1 (GraphPad Software, San Diego, CA, USA). Two-way ANOVA was performed (treatment × subregion) with a Tukey’s *post hoc* test for all comparisons. The Shapiro–Wilk and Kolmogorov–Smirnov were performed to assess normality. Statistical significance was assessed using an alpha level of 0.05. Data are presented as intercleaved box and whisker plots with minimum and maximum values and individual plot points for each data set. All statistical comparisons (Appendix A) and all raw data (represented as mean ± SEM; Appendix A) have been included in the supplemental section, regardless of significance.

## 3. Results

### 3.1. Blood EtOH Concentrations (BECs)

Blood EtOH concentrations were obtained 60 min after EtOH administration at two timepoints: after the first and last dose. We were unable to obtain sufficient blood from one animal when collecting samples at the first timepoint, therefore we had n = 5 for the first time point and n = 6 samples for the second timepoint. Results show that 60 min after EtOH administration, BECs were (mean ± SEM, mg/dL) = 139.21 ± 9.46 and 119.39 ± 2.14 after the first and last dose, respectively. These BECs are consistent with the ranges seen within previous human adolescent drinking analyses [53].

### 3.2. AIE Induced Changes in PFC Astrocyte Morphology and PAP-Synaptic Proximity in a Subregion-Dependent Manner, but Only after a Period of Forced Abstinence

Before assessing AIE-induced changes in PAP-synaptic interactions, we measured changes in astrocyte morphology based on whole-cell volume. Using Imaris, we quantified the volume of individual Lck-GFP^+^ astrocytes from the ACC, mPFC (IL and PL), LO-OFC, and VO-OFC (Figure 2A–D) 24 h after the last dose (i.e., PND 46) and after a 26-day forced abstinence period (i.e., PND 72). There was no treatment (AIE) effect on astrocyte volume during peak withdrawal (F (1, 476) = 1.251, *p* = 0.2639; Figure 2A,B) or after the 26-day forced abstinence period (F (1, 471) = 0.1942, *p* = 0.6597; Figure 2C,D). However, there was a significant subregion effect at both timepoints ((F (1, 476) = 84.85, *p* < 0.001 and F (1, 471) = 208.4, *p* < 0.001, respectively). When we compared the generalized characteristics of astrocyte volume in EtOH-naïve animals (i.e., comparison of H_2_O controls across subregions), we found that there were distinct differences in astrocyte volume that were dependent on the PFC subregion. At PND 46, ACC and mPFC had the lowest astrocyte volume when compared with LO-OFC and VO-OFC (*p*’s < 0.0001; Figure 3A,B). There was no significant difference between ACC and mPFC astrocyte volume (*p* = 0.1199) or LO-OFC and VO-OFC astrocyte volume (*p* > 0.9999). Interestingly, by adulthood (PND 72), the subregion-dependent effect on astrocyte volume became more refined (F (1, 471) = 208.4, *p* < 0.001). At the adult timepoint, a significant difference emerged between ACC astrocyte volume and mPFC astrocyte volume, with mPFC emerging as having the larger astrocyte volume (*p* = 0.0349). As demonstrated at PND46, LO-OFC and VO-OFC astrocytes remained significantly smaller than both ACC (*p* < 0.0001 and *p* < 0.0001, respectively) and mPFC (*p* < 0.0001 and *p* < 0.0001, respectively) astrocytes. There was no significant difference between astrocyte volume when comparing LO-OFC and VO-OFC (*p* = 0.1322).

To determine the effects of AIE on PAP-synaptic proximity, we used immunohistochemistry to probe for the postsynaptic density marker, PSD-95, in tissue along with Lck-GFP^+^ astrocytes (Figure 3A–C). There was a significant treatment effect (AIE), subregion effect, and interaction on PAP-synaptic co-localization during peak withdrawal (24 h after the last dose) (treatment effect: F (1, 476) = 5.097, *p* = 0.0244; subregion effect: F (1, 476) = 160.5, *p* < 0.0001; interaction: F (1, 476) = 3.480, *p* = 0.0159; Figure 3A,B). However, upon *post hoc* analysis, there were no significant effects of AIE within subregions when compared to controls. When we compared the generalized characteristics of astrocyte PAP-synaptic proximity in EtOH-naïve animals (i.e., comparison of H_2_O controls across subregions) at this early timepoint, there was no significant difference between ACC and mPFC co-localization (*p* > 0.9999). There were significantly more PAPs co-localizing with synapses in LO-OFC and VO-OFC subregions when compared to ACC (*p* < 0.0001 and *p* < 0.0001, respectively) and mPFC (*p* < 0.0001 and *p* < 0.0001, respectively). There were significantly more PAPs co-localizing with synapses within the VO-OFC when compared with LO-OFC (*p* < 0.0001). By adulthood (PND 72), the subregion-dependent effect of PAP-synaptic co-localization became more refined. PAP-synaptic co-localization was significantly lower in mPFC than ACC (*p* < 0.0001), LO-OFC (*p* < 0.0001), and VO-OFC (*p* < 0.0001). PAP-synaptic co-localization was significantly higher in LO-OFC (*p* < 0.0001) and VO-OFC (*p* < 0.0001) subregions when compared to ACC, while VO-OFC had significantly more PAP-synaptic co-localization than LO-OFC (*p* < 0.0001).

After the 26-day forced abstinence period (PND 72) there was a robust treatment effect (AIE), subregion effect, and interaction on PAP-synaptic co-localization (treatment effect: F (1, 502) = 35.35, *p* < 0.0001; subregion effect: F (1, 502) = 237.6, *p* < 0.0001; interaction: F (1, 502) = 24.70, *p* < 0.0001; Figure 3C,D). When we compared the generalized characteristics of astrocyte PAP-synaptic proximity in EtOH-naïve animals (i.e., comparison of H_2_O controls across subregions) at this adult timepoint, there was significantly less PAP-synaptic co-localization in the mPFC when compared to ACC (*p* < 0.0001), LO-OFC (*p* < 0.0001), and VO-OFC (*p* < 0.0001). Once again, PAP-synaptic co-localization was significantly higher in LO-OFC (*p* < 0.0001) and VO-OFC (*p* < 0.0001) subregions when compared to ACC. However, at this adult timepoint, we began to see a differentiation between OFC subregions. Treatment-naïve astrocytes within the VO-OFC showed significantly higher PAP-synaptic co-localization when compared to LO-OFC astrocytes (*p* < 0.0001).

*Post hoc* analysis of the treatment effect following forced abstinence demonstrated a significant decrease in PAP-synaptic co-localization in the ACC (Figure 3C,D) and VO-OFC (*p* < 0.0001; Figure 3C,D). There was no effect of AIE on PAP-synaptic co-localization within the mPFC (*p* = 0.9444) or the LO-OFC (*p* = 0.9974).

To rule out the loss of synapses as a driver in the loss of PAP-synaptic interactions in the ACC and VO-OFC, we quantified PSD-95 protein expression in all prefrontal subregions during peak withdrawal (PND 46) and following the 26-day forced abstinence period (PND 72). We used the Imaris ‘Spot’ function to count the number of individual PSD-95 puncta within the ROI in which our Lck-GFP^+^ astrocytes were imaged. We found no effect of AIE on PSD-95 expression in the ACC, mPFC (PL and IL), LO-OFC, or VO-OFC at PND 72 (*p* > 0.05; Appendix A) when loss of PAP-synaptic coupling occurs.

### 3.3. AIE Results in Cortical Subregion Dependent Shifts in Dendritic Spine Maturation after Forced Abstinence

PAP-synaptic proximity is critical for appropriately targeted astrocyte contact-mediated and astrocyte-secreted signaling, particularly with regard to regulating dendritic spine maturation [41]. We wanted to determine if the AIE-induced PAP-synaptic loss of proximity after the 26-day forced abstinence period was associated with changes in dendritic spine maturation and density [56,57]. Since the AIE-induced effects on PAP-synaptic co-localization were only observed in adulthood, and to avoid any potential confounding withdrawal effects, we focused on the adult timepoint (after the 26-day forced abstinence period) for all remaining experiments. There was no significant effect of subregion on spine density (F (1, 352) = 0.7979, *p* = 0.3723; Appendix A). There was a significant effect of treatment on spine density (F (3, 352) = 3.764, *p* = 0.011; Appendix A); however, *post hoc* analysis revealed no significant treatment effects when comparing AIE to control within a subregion (*p* > 0.05; Appendix A). The lack of AIE-induced change in dendritic spine density is consistent with the quantification of PSD-95, an indicator of synaptic number. The absence of changes in dendritic spine density and PSD-95 expression suggests that synaptic loss is not driving the AIE-induced changes in PAP-synaptic interactions.

Based on morphological measures of spine neck length and spine head width (Figure 4A–C), we observed no effect of treatment (AIE) (F (3, 86) = 3.368, *p* = 0.0699), subregion (F (3, 86) = 1.331, *p* = 0.2696), or interaction (F (3, 86) = 1.188, *p* = 0.3192) on the % of immature (i.e., filopodia) spines (Figure 4D). When comparing the % of intermediate (i.e., long/thin) spines, there was a significant effect of treatment (F (3, 440) = 4.040, *p* = 0.0450) and subregion (F (3, 440) = 12.04, *p* < 0.0001) and a significant interaction (F (3, 440) = 8.833, *p* < 0.0001). When subregion differences in dendritic spine type were compared in EtOH-naïve animals (i.e., comparison of H_2_O controls across subregions), there was a significantly higher % of intermediate spines within the LO-OFC (when compared with the ACC (*p* < 0.0001; Figure 4D), mPFC (*p* < 0.0001; Figure 4E), and VO-OFC (*p* < 0.0001; Figure 4E). There were no differences in the % of intermediate spines when comparing all other subregions (*p* > 0.05, Figure 4E). *Post hoc* analysis also revealed that there were no AIE-induced changes in the % of intermediate spines in the ACC (*p* = 0.5438) and mPFC (*p* = 0.2931); however, there was a significant AIE-induced decrease in the number of intermediate spines within the LO-OFC (*p* = 0.0226; Figure 4E) and an increase in the number of intermediate spines within the VO-OFC (Figure 4E).

Assessment of the % of mature (i.e., mushroom) dendritic spines following AIE treatment and within/across subregions revealed a subregion effect (F (3, 390) = 11.32, *p* < 0.0001), no treatment effect (F (1, 390) = 1.530, *p* = 0.2168), and an interaction between the two measures (F (3, 390) = 8.043, *p* < 0.0001). *Post hoc* analysis revealed that when subregion differences in dendritic spine type were compared in EtOH-naïve animals (i.e., comparison of H_2_O controls across subregions) there were significantly less mature spines within the LO-OFC when compared to the ACC (*p* < 0.0001; Figure 4D), mPFC (*p* < 0.0001; Figure 4F), and VO-OFC (*p* = 0.0003; Figure 4F). Interestingly, when assessing the effects of treatment, there was no AIE-induced change in the % of mature dendritic spines in the ACC (*p* = 0.6307) or mPFC (*p* = 0.9996). However, complementary to the changes in intermediate spines, there was a significant increase in the % of mature spines within the LO-OFC (*p* = 0.0403) and a significant decrease in the % of mature spines within the VO-OFC (*p* = 0.0079).

### 3.4. AIE-Induced loss of PAP-Synaptic Co-localization Is Not Driven by Changes in Expression of Synaptic Stabilization Proteins

To determine if the loss of PAP-synaptic proximity in adulthood was due to a loss of bridging proteins involved in tripartite synapse stabilization, we investigated changes in the expression of neuroligins 1 and 3 and their presynaptic partner, neurexin. Analysis of these critical stabilizing proteins revealed no AIE-induced changes in the expression in any of the PFC subregions, indicating that a loss of these proteins is not a driver of the AIE-induced PAP-synaptic decoupling (Appendix A). To further explore the effects of AIE on neurexin-neuroligin interactions at the tripartite synapse, we assessed the co-localization of neurexin-neuroligin 1 and neurexin-neuroligin 3.

Analysis of neurexin-neuroligin 1 co-localization revealed a significant treatment effect (F (3, 286) = 14.08, *p* = 0.0002), subregion effect (F (3, 286) = 48.01, *p* < 0.0001), and a significant interaction (F (3, 286) = 7.137, *p* = 0.0001). When subregion differences in neurexin-neuroligin 1 co-localization were compared in EtOH-naïve animals (i.e., comparison of H_2_O controls across subregions), there was significantly higher co-localization within the ACC when compared to mPFC (*p* < 0.0001; Figure 5B), LO-OFC (*p* = 0.0009; Figure 5B), and VO-OFC (*p* < 0.0001; Figure 5B), with VO-OFC having the lowest neurexin-neuroligin 1 co-localization of all 4 subregions. *Post hoc* analysis also revealed an effect of AIE on two PFC subregions. Specifically, AIE resulted in a significant decrease in neurexin-neuroligin 1 co-localization in the ACC (*p* = 0.0002; Figure 5B) and within the VO-OFC (*p* = 0.0394), consistent with the loss of PAP-synaptic coupling within these two specific subregions. There was no AIE-induced change in neurexin-neuroligin 1 co-localization within the mPFC (*p* = 0.9979; Figure 5B) or the LO-OFC (*p* = 0.9392; Figure 5B).

Analysis of neurexin-neuroligin 3 co-localization revealed a significant treatment effect (F (3, 286) = 6.664, *p* = 0.0103), subregion effect (F (3, 286) = 19.97, *p* < 0.0001), and a significant interaction (F (3, 286) = 7.874, *p* < 0.0001). When subregion differences in neurexin-neuroligin 3 co-localization were compared in EtOH-naïve animals (i.e., comparison of H_2_O controls across subregions), there was significantly higher co-localization within the ACC when compared to mPFC (*p* < 0.0001; Figure 6B) and VO-OFC (*p* < 0.0001; Figure 6B), but not when compared to the LO-OFC (*p* = 0.5565; Figure 6B). As with neurexin-neuroligin 1, co-localization of neurexin-neuroligin 3 was the highest within the ACC and lowest within the VO-OFC when all subregions were compared. Further *post hoc* analysis revealed an effect of AIE on two PFC subregions: specifically, AIE resulted in a significant decrease in neurexin-neuroligin 3 co-localization in the ACC (*p* = 0.00184; Figure 6B) and within the VO-OFC (*p* = 0.0635; Figure 6B), once again consistent with the loss of PAP-synaptic coupling within these two specific subregions. There were no AIE-induced changes in neurexin-neuroligin 3 co-localization within the mPFC (*p* = 0.2768; Figure 6B) or the LO-OFC (*p* > 0.9999; Figure 6B).

## 4. Discussion

The aim of this study was to investigate the impact of adolescent EtOH exposure on astrocyte morphology, PAP-synaptic proximity, and dendritic spine maturation in different PFC subregions, as well as to determine whether loss of neurexin- neuroligin cell adhesion proteins were a driving factor in the changes observed. Overall, we found a cortical subregion-specific loss of PAP-synaptic co-localization/proximity after a 26-day forced abstinence period (in adulthood) within the ACC and VO-OFC that were not present during acute withdrawal. AIE-induced loss of PAP-synaptic co-localization was not due to an overall reduction in astrocyte volume or a loss of synapses; however, the loss of PAP-synaptic proximity did correlate with a significant shift towards a less mature dendritic spine phenotype in adulthood within the VO-OFC, as illustrated by an increase in intermediate spines and a corresponding decrease in mature spines (not seen in other subregions). Lastly, despite no changes in neurexin, neuroligin 1, or neuroligin 3 protein expression following AIE, there was a reduction in neuroligin-neurexin co-localization in the ACC and VO-OFC that correlated with a loss of PAP-synaptic co-localization and the VO-OFC-specific loss of dendritic spine maturity. Additional analysis demonstrated robust differences in AIE-naïve animals across PFC subregions, which included baseline differences in astrocyte volume, PAP-synaptic co-localization, dendritic spine morphology, and neurexin-neuroligin co-localization.

The human and rodent PFC continues to undergo structural and functional neuronal refinement during adolescence [58,59,60,61,62,63,64]. This ongoing developmental period is suggested to be a time of increased vulnerability to AIE-induced neuronal disruption. This has been demonstrated by Broadwater et al. (2018) in which they showed that AIE perturbed resting state connectivity using functional MRI connectivity between OFC-striatum and OFC-nucleus accumbens in rats [65]. These findings correlate with multiple studies demonstrating AIE-induced deficits in PFC-dependent behaviors such as behavioral flexibility, increased disinhibition, and increased propensity to self-administer EtOH; all of which persist into adulthood [66,67,68,69]. However, there is growing evidence that behavioral outputs are highly reliant on astrocyte modulation of neuronal function. This has been demonstrated in a number of studies using designer receptors exclusively activated by designer drugs (DREADDs) that specifically target astrocytes and result in subsequent modulation of rodent behavior (for a review, see Hwang et al. [70]). One study of particular importance was conducted by Erickson and colleagues [35] in which they showed that activation of astrocyte-specific excitatory DREADDs within the PFC (IL, PL, and ACC combined) can regulate EtOH consumption, supporting a critical role for astrocytes in addiction-related behavior.

Astrocytes are able to modulate neuronal function and thus behavior via the uptake of neurotransmitter from the synaptic cleft, the release of gliotransmitters, and through contact-mediated and astrocyte-secreted signaling factors (for a review, see Allen [71] and Blanco-Suarez et al. [72]). One structural characteristic that all of these signaling mechanisms require is astrocyte proximity to the synapse. In the current study, we found that the ACC and VO-OFC appear to be particularly vulnerable to disruption of PAP-synaptic co-localization, suggesting a loss of astrocyte proximity to the synapse. Despite assessing this measure acutely within 24 h of the last dose, the loss of PAP-synaptic coupling only emerged in adulthood after prolonged abstinence, and not in all subregions of the PFC. It is unclear whether these subregion differences are due to astrocyte heterogeneity or variations in the state of developmental maturation of the PAP-synaptic relationship, within the different PFC subregions. However, there are indications from our data that suggest that there could indeed be very different astrocyte populations within these subregions. For example, in every comparative statistical measure conducted in this study assessing differences between EtOH naïve animals (i.e., control animals only) in different subregions, we were able to demonstrate that the ACC has significantly higher astrocyte volume with less PAP-synaptic coupling, a higher ratio of mature spines, and higher levels of neurexin-neuroligin co-localization than all other subregions, particularly when compared to the VO-OFC. While there were a variety of differences across the various subregions at baseline, the most pronounced differences were seen when comparing ACC and VO-OFC. Interestingly, these two regions were also the most vulnerable to PAP-synaptic decoupling. Why the loss of PAP-synaptic proximity occurred specifically within the VO-OFC and the ACC requires further investigation, as does the mechanism underlying the delayed occurrence of this phenomenon (i.e., not present at the acute timepoint but emerges during prolonged abstinence, in adulthood). Understanding astrocyte heterogeneity across PFC subregions and the unique neuronal populations that these astrocytes serve will be necessary moving forward.

Due to the AIE-induced loss of PAP-synaptic co-localization in ACC and VO-OFC we investigated the impact that the loss of proximity would have on dendritic spine number and morphology. Previous work by Witcher et al., 2007 [42] using serial section electron microscopy of mature rat hippocampal slices demonstrated that synapses are larger when PAPs are present. This is consistent with the findings in the current study in which we demonstrated a shift towards a less mature dendritic spine phenotype that only occurred in the VO-OFC where there was a significant loss of PAP-synaptic proximity, suggesting a strong relationship between dendritic spine maturity and PAP proximity. The current finding that AIE increased the presence of intermediate dendritic spines in the VO-OFC is consistent with McGuier et al., 2015 [73] who previously showed that the more immature phenotype is present after a 7-day abstinence period but not during peak withdrawal. However, here we showed that even after a prolonged period (26 days) of abstinence, the changes in dendritic spine phenotype persist.

An interesting finding that emerged from the dendritic spine analysis was that, following EtOH exposure, when the % of mature dendritic spines decreased in the VO-OFC, they increased in the LO-OFC. When the % of intermediate dendritic spines increased in the VO-OFC, they decreased in the LO-OFC without affecting overall spine density. It is possible that the increase in mature spines within the LO-OFC was due to an increase in PAP-synaptic proximity without impacting the overall number of colocalized puncta, though this is difficult to determine due to the limitations of the analysis in which we used a threshold of 0.5 µM to define proximity (due to resolution limits). We are currently unable to determine with any accuracy whether PAPs are in fact getting closer to the synapses as a result of AIE exposure. Further analysis using super-resolution and/or electron microscopy would be beneficial to further define changes in proximity across PFC subregions. In summary, there were clear differences in sensitivity to AIE-induced loss of PAP-synaptic proximity and corresponding dendritic spine morphology across OFC subregions, despite no change in overall spine density.

Previous work in mice has shown that proteins responsible for bridging the tripartite synapse, such as neurexin-neuroligin and eph-ephrin, are critical for dendritic spine stabilization and maturation [40], synaptic function, and even astrocyte complexity [43]. Since AIE resulted in a sub-region dependent shift towards a less mature dendritic spine phenotype, we investigated if a loss of neurexin and neuroligin expression could be driving the changes in dendritic spine phenotype. Following AIE, there were no changes in expression of neurexin, neuroligin 1, or neuroligin 3. However, work by Ko et al., 2009 [44] previously showed that using mutational disruption of neurexin-neuroligin binding prevents normal increases in pre- and post-synaptic size. Based on this finding, we next assessed co-localization of neurexin-neuroligin 1 and neuroxin-neuroligin 3, hypothesizing that the loss of PAP-synaptic proximity may prevent appropriate localization of neurexin-neuroligins and therefore impede normal synaptic enlargement. In the current study, we were able to confirm that, despite no change in the expression of either neurexin or neuroligins 1 or 3 following AIE, there was a decrease in co-localization of neurexin-neuroligin 1 and 3 specifically in the ACC and VO-OFC (where a change in dendritic spine phenotype was also observed). However, this finding is inconclusive since it is possible that AIE directly disrupts neurexin-neuroligin binding sites or drives the generation of alternate splice variants, thereby disrupting the ability of neurexin-neuroligin to interact. Another possibility is that leucine-rich repeat transmembrane neuronal proteins (LRRTM2), which bind to neurexins and work synergistically with neuroligin 1 are impacted by AIE, thus modulating neurexin-neuroligin 1 interactions [74]. However, based on the pro-synaptogenic properties of LRRTM2 [75,76] we would likely see changes in PSD-95 puncta or changes in dendritic spine density, indicative of pro or anti-synaptogenic signaling, depending on the type of AIE-induced protein modulation that occurs. Further work is necessary to determine whether neurexin-neuroligin binding is impaired and if so, precedes PAP-synaptic decoupling.

Changes in astrocyte morphology and increased protein expression at the level of GFAP antibody staining are considered to be signs of astrocyte reactivity/activation and correlated with the induction of neuroinflammation [77,78,79] and injury. GFAP upregulation and immune activation have also been demonstrated to occur following EtOH exposure in rats and mice [7,80,81]. However, GFAP antibody staining only captures the backbone of the astrocyte and not the fine processes that interact with blood vessels, other astrocytes, or synapses. Moreover, we have previously demonstrated that hippocampal astrocytes begin releasing factors associated with reactivity much earlier than the appearance of increased GFAP expression [52] in an identical model, suggesting that GFAP expression is not the most sensitive measure for astrocyte reactivity. However, there is no doubt that astrocytes become reactive and can drive neuroinflammation or the response to neuroinflammation [82,83,84]. The questions that remain are whether changes in GFAP expression are reflective of changes at the PAPs, and whether GFAP expression can be correlated with a loss of PAP-synaptic coupling. Furthermore, it has yet to be demonstrated that neuroimmune activation can directly drive PAP-synaptic decoupling.

We have demonstrated that AIE-induced loss of PAP-synaptic coupling is PFC subregion-dependent. Based on astrocyte volumetric measures, it is quite possible that we are comparing two different astrocyte populations that are likely providing support for different types of neurons. This raises questions of whether all synapses are equally dependent on astrocyte support for spine morphology, and whether all tripartite synapses are equally dependent on the presence of these particular NLGN-NRXN interactions.

However, the implications of AIE-induced loss of PAP-synaptic coupling go beyond the health of dendritic spines and could significantly impact the glymphatic system and astrocyte-astrocyte communication, both of which are critical for maintaining the movement of cerebrospinal fluid and ion homeostasis. Previous work demonstrates that even chronic 30-day exposure to 1.5 g/kg EtOH (i.p.) can significantly diminish glymphatic function in the mouse [85] and has been associated with the development of dementia in humans [86], particularly in heavy drinkers [87].

## 5. Conclusions

These data provide novel insight into the selective vulnerability of astrocyte-synaptic interactions and dendritic spine morphology within PFC subregions following AIE. Interestingly, many of these effects occur well after the acute effects of EtOH have dissipated, suggesting delayed and enduring changes to astrocyte-synaptic interactions that persist into adulthood. Given the importance of astrocytes in regulating synaptic activity and their emerging role in behavioral regulation, further work is necessary to determine the molecular mechanisms that drive the loss of PAP-synaptic coupling and to understand why some PFC subregions display increased vulnerability to the effects of AIE. How changes in PAP-synaptic proximity and dendritic spine morphology contribute to the neuronal and behavioral changes observed following repeated AIE will be an important step moving forward and may provide novel non-neuronal targets for future pharmacological interventions.

## Figures and Tables

**Figure 1 cells-11-03111-f001:**
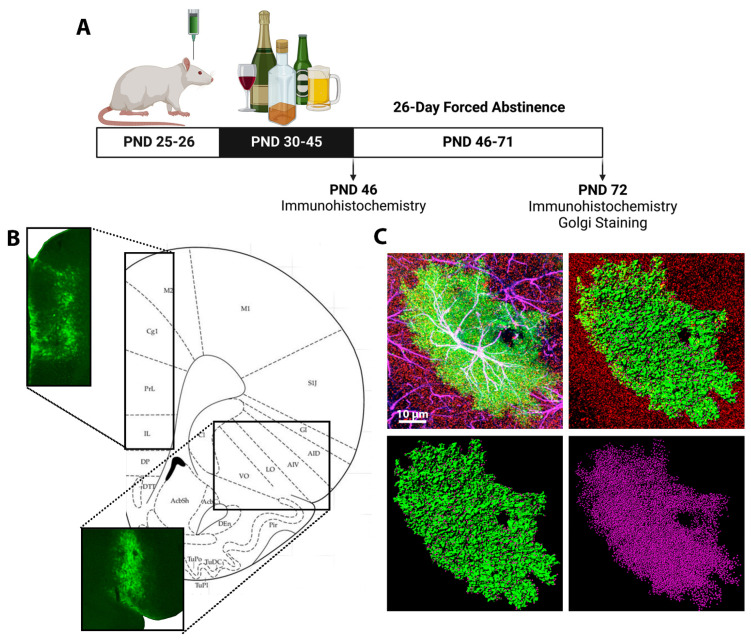
Experimental Design: (**A**) Animals received intracranial injections of an astrocyte-specific adeno-associated virus directly into the ACC, mPFC, or OFC on PND 25–26. Beginning PND 30, animals received 5 g/kg of EtOH or water via oral gavage on an intermittent, 2 days on, 1 day off, 2 days on, 2 days off, schedule. Tissue was collected on PND 46, during peak withdrawal, or after a 26-day forced abstinence period, on PND 72. (**B**) Validation of Lck-GFP expression in the ACC and mPFC (upper left image) and the OFC (bottom image). (**C**) Representative images of AAV+ astrocyte imaging with confocal microscopy and reconstruction using Imaris. The top left image is a confocal image of an AAV+ astrocyte (green), with GFAP (magenta), co-localization of AAV+ and GFAP (white), and PSD-95 (red), scale bar, 10 µm. The top right image is a reconstructed astrocyte (green) and PSD-95 (red). The bottom left image is a reconstructed surface rendered astrocyte (green) with PSD-95 (magenta) within 0.5 µm of the astrocyte. The bottom right image is isolated PSD-95 (magenta) that is colocalized with the astrocyte of interest.

**Figure 2 cells-11-03111-f002:**
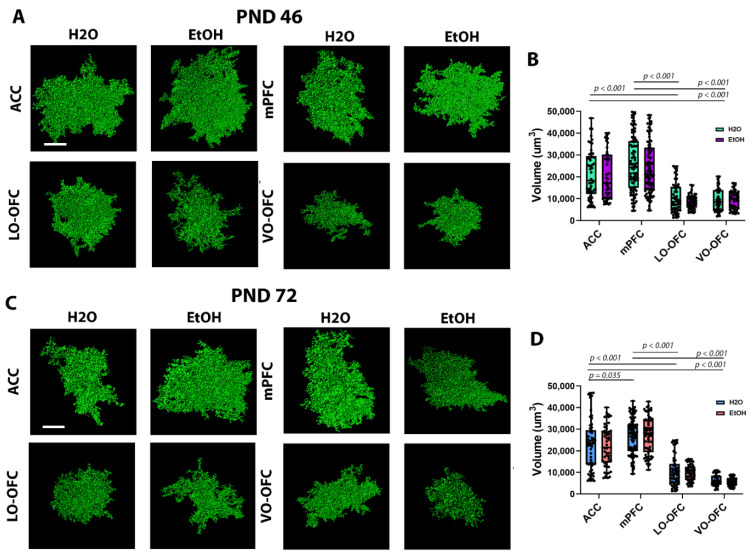
Single cell imaging and analysis of astrocyte volumes 24 h after AIE during withdrawal and 26-days after AIE. (**A**) Representative images of astrocyte surface rendering from the ACC, mPFC, LO-OFC, and VO-OFC 24 h after the final dose of EtOH. Scale bar, 20 µm. (**B**) Quantification of astrocyte volumes 24 h after the final dose of EtOH. When comparing astrocyte volume across subregions within the control groups, there was a significant decrease in astrocyte volume in LO-OFC and VO-OFC when compared to ACC and mPFC. There was no significant treatment effect (H_2_O vs. EtOH) within subregions. (**C**) Representative images of astrocyte surface rendering from the ACC, mPFC, LO-OFC, and VO-OFC following a 26-day forced abstinence period. Scale bar 20 µm. (**D**) Quantification of astrocyte volumes following a 26 day-day forced abstinence period. When comparing control conditions across subregions, there was a significant decrease in astrocyte volume in LO-OFC and VO-OFC when compared to ACC and mPFC. There was no significant treatment effect (H_2_O vs. EtOH) within subregions (*p* > 0.05). Error bars represent minimum and maximum values in each data set. Data presented as box and whisker plots with interquartile range with mean, minimum, and maximum measures indicated along with individual data points. Analysis: two-way ANOVA with Tukey’s *post hoc* comparison, n = 5/treatment group.

**Figure 3 cells-11-03111-f003:**
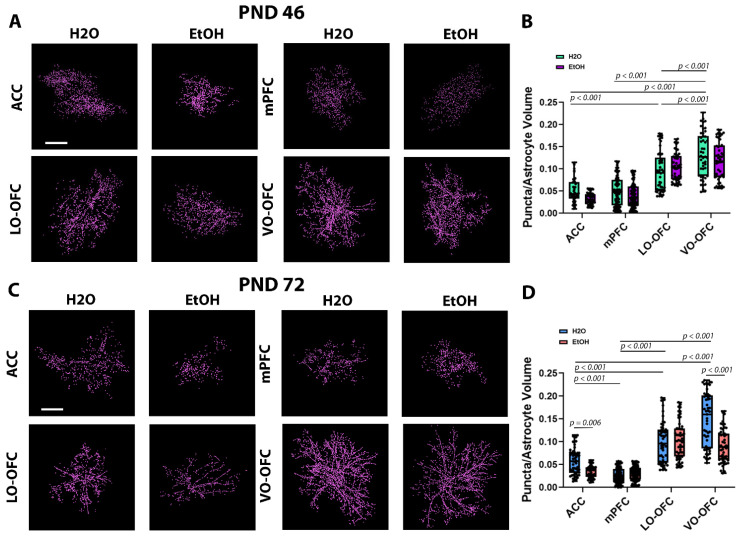
Single astrocyte co-localization with PSD-95 and analysis 24 h (PND 46) and 26-days (PND 72) after AIE. (**A**) Representative images of PSD-95 colocalized within 0.5 µm of a Lck-GFP^+^ astrocyte in the ACC, mPFC (IL and PL), LO-OFC, and VO-OFC 24 h after AIE. Scale bar 20 µm. (**B**) Quantification of puncta/astrocyte volume (puncta/µm^3^) in each of the PFC subregions. When comparing puncta number across subregions within the control groups, there was a significant increase in puncta number/astrocyte volume in LO-OFC and VO-OFC when compared to ACC and mPFC. There was no significant treatment effect on puncta number (H_2_O vs. EtOH) within subregions. (**C**) Representative images of PSD-95 colocalized within 0.5 µm of an AAV+ astrocyte in the ACC, mPFC (IL and PL), LO-OFC, and VO-OFC 26-days after AIE. Scale bar 20 µm. (**D**). When comparing puncta number across subregions within the control groups, there was a significant increase in puncta number/astrocyte volume in LO-OFC and VO-OFC when compared to ACC and mPFC. There was a significant treatment effect on puncta number (H_2_O vs. EtOH) demonstrated by a decrease in astrocyte-synapse co-localization in the ACC (*p* < 0.006) and the VO-OFC (*p* < 0.001). Data presented as box and whisker plots with interquartile range with mean, minimum, and maximum measures indicated along with individual data points. Analysis: two-way ANOVA with Tukey’s *post hoc* comparison, n = 5/treatment group.

**Figure 4 cells-11-03111-f004:**
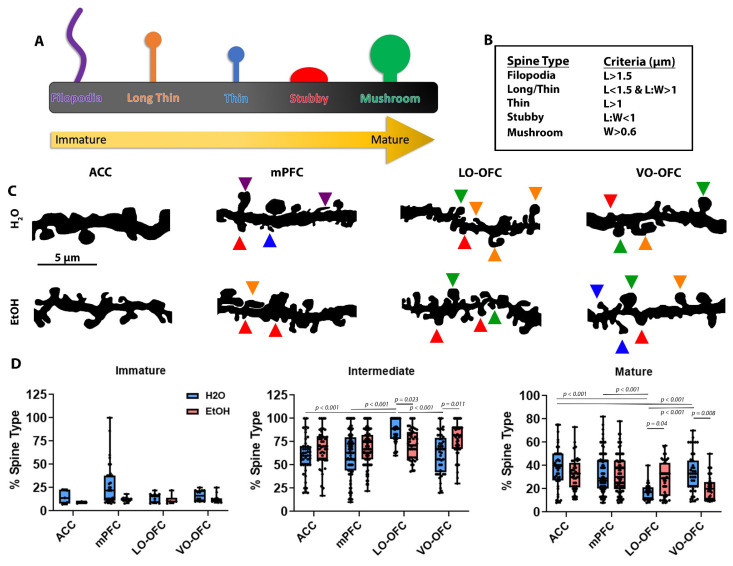
The effects of AIE on dendritic spine maturation after a 26-day forced abstinence period. (**A**) Cartoon representation of the classification of different dendritic spine types and how they correlate to maturation. (**B**) Table depicting how dendritic spines are classified. (**C**) Representative images of secondary dendrites depicting spine types in the ACC, mPFC (IL/PL), LO-OFC, and VO-OFC. Triangles indicate spine type based on color: Filopodia—purple; Long Thin—orange; Thin—blue; Stubby—red; Mushroom—green. (**D**) Analysis of dendritic spine maturation in the ACC, mPFC (IL and PL), LO-OFC, and VO-OFC. There was no change in the % of immature dendritic spines in any subregion (*p* > 0.05) following AIE or when comparing subregions (*p* > 0.05). In the LO-OFC, there was a significant decrease in intermediate spines (*p* = 0.023) and an increase in mature spines (*p* = 0.04) following AIE. In the VO-OFC, there was a significant increase in intermediate spines (*p* = 0.011) and a significant decrease in mature spines (*p* = 0.008) following AIE. Data presented as box and whisker plots with interquartile range with mean, minimum, and maximum measures indicated along with individual data points. Average spine densities were calculated using 2–3 separate dendrites from at least 4–5 separate image stacks per animal for a total of 15 dendritic branches/animal (n = 3/treatment group). Analysis: two-way ANOVA with Tukey’s *post hoc* comparison.

**Figure 5 cells-11-03111-f005:**
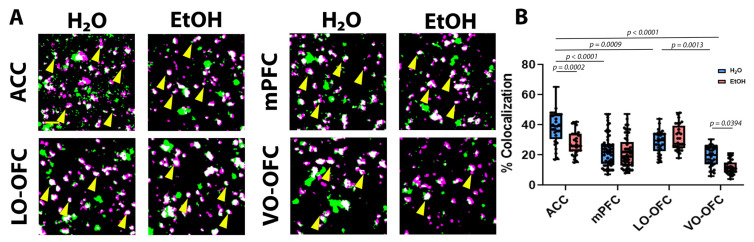
The effect of AIE on neurexin-neuroligin 1 interactions after a 26-day forced abstinence period. We found that AIE did not change neurexin or neuroligin 1 expression following a 26-day forced abstinence period (see Appendix A). (**A**) Representative images of neurexin (green) and neuroligin 1 (magenta) in the ACC, mPFC (IL and PL), LO-OFC, and VO-OFC. Scale bar, 2 µm. (**B**) Quantification of neurexin and neuroligin 1 co-localization. There was a significant decrease in neurexin-neuroligin 1 co-localization in the ACC (*p* = 0.0002) and the VO-OFC (*p* = 0.0394) following AIE. Data presented as box and whisker plots with interquartile range with mean, minimum, and maximum measures indicated along with individual data points. Three 5 µm image stacks from randomly selected hemispheres were captured from 3 separate brain slices per animal (n = 9–10 brains/treatment group). Analysis: two-way ANOVA with Tukey’s *post hoc* comparison.

**Figure 6 cells-11-03111-f006:**
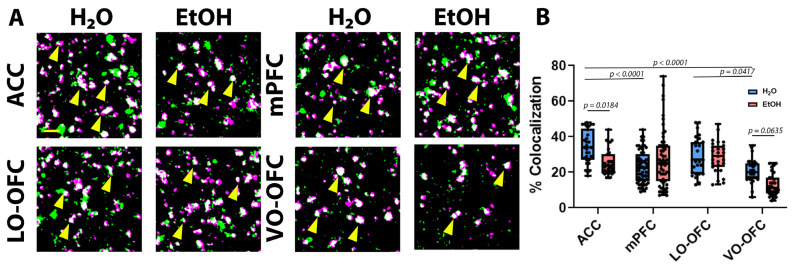
The effect of AIE on neurexin-neuroligin 3 interactions after a 26-day forced abstinence period. We found that AIE did not change neurexin or neuroligin 3 expression following a 26-day forced abstinence period (see Appendix A). (**A**) Representative images of neurexin (green) and neuroligin 3 (magenta) in the ACC, mPFC (IL and PL), LO-OFC, and VO-OFC. Scale bar, 2 µm. (**B**) Quantification of neurexin and neuroligin 3 co-localization. There was a significant decrease in neurexin-neuroligin 3 co-localization in the ACC (*p* = 0.0184) and the VO-OFC (*p* = 0.0635). Data presented as box and whisker plots with interquartile range with mean, minimum, and maximum measures indicated along with individual data points. Three 5 µm image stacks from randomly selected hemispheres were captured from 3 separate brain slices per animal (n = 9–10 brains/treatment group). Analysis: two-way ANOVA with Tukey’s *post hoc* comparison.

## Data Availability

The data presented in this study is the property of the US Federal Government, however, are still available on request from the corresponding author.

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
