# Peer review of "Diverging Effects of Adolescent Ethanol Exposure on Tripartite Synaptic Development across Prefrontal Cortex Subregions"

_cells, 2022, doi:10.3390/cells11193111_

Round 1

Reviewer 1 Report

In this paper, the authors aim to study the effects of binge ethanol exposure on astrocyte morphology, proximity of peripheral astrocytic processes (PAP) to synapses, dendritic spine maturation, and to determine whether loss of neuroligin-neurexin cell adhesion proteins are involved. They used a model of male Sprague-Dawley rats from postnatal day 24 to 72. The authors found no change in astrocytic volume, but they reported changes in PAP synaptic proximity and in dendritic spine maturation in key brain areas including the anterior cingulate cortex (ACC) and in orbito frontal ventral (VO-OFC) cortex. In addition, changes in the co-localization (but not in expression) of neurexin-neuroligin1/3 were reported in the same areas.

The topic is of high interest and even epidemiological importance, due to the well-known negative effects of excessive and poorly controlled alcohol consumption in adolescents and young adults. Apart from social and behavioral problems associated with alcohol, other direct effects on the brain and brain development are needed to be addressed. Astrocytes are fundamental for brain homeostasis regulation and, thus, is important to understand the effects of ethanol in the function of these glial cells. This is further underscored when ethanol acts on the brains of adolescents, which are at a critical period of development. The paper addresses some of these questions, providing new observations that may hep to understand the effects of binge drinking in astrocytes of adolescents. This research can contribute to the field and therefore, the paper can be considered for publication. Figures 1 and 4 have nice schematics that help the reader to understand the experiments conducted. However, there are several issues that need to be solved first. The following comments and suggestions are aimed to help authors improve their manuscript.

Comments:

-          In lines 35-36, the paper by Patrick and Schulenberg is specific to adolescents in the USA. Please add this information to the phrase.

-          In lines 37-39, I suggest replacing the reference with the original paper (Pubmed number PMID: 31912524), as the included reference is focused on alcohol-related deaths during Covid-19 pandemic (although it is correctly used to reference the phrase at lines 41-43).

-          In lines 39-41, the WHO citation is missing from the References section of the paper. Please include this reference and double check if any other is missing.

-          In line 44, the word adolescents should be replaced with adolescence.

-          In line 54, Koob & Volkow citation is missing from the references section.

-          Lines 56-57. Please correct this phrase.

-          Line 64. The dot should be substituted by a comma.

-          Lines 68-71. Please check the references. The terms infralimbic or prelimbic are not mentioned in the paper of Warren et al. (2016). It should be substituted for a more adequate reference.

-          Lines 108-110. Please check the numbers of single astrocyte interactions. The paper by Bushong et. Al (2002), mentions “an astrocyte occupying 66,000 μm3 of neuropil would oversee 140,000 synapses”, the number up to 100,000 is not mentioned in that paper. In addition, the paper of Bushong doesn´t mention 2 million synapses in humans, as this information was reported subsequently in the paper of Oberheim et al (2009) (PMID: 19279265), which should be cited in this phrase as well.

-          Following the recommendations by ARRIVE 2.0 guidelines, SyRCLE and similar initiatives, I kindly suggest the authors to provide some additional information about the experimental procedures.

o   Animal and surgical procedures: Please indicate if there was an acclimatization/habituation period of the animals before the start of the experiments. Please state if the animals were randomized into the experimental groups (and how was this randomization performed, also if the person doing this randomization or animal group distribution was blinded or not). Please add a clearer and more explicit description of the control and experimental groups used, indicating the number of animals per phase of the experimental procedure. For example, if you planned 5 animals per group and had 4 groups, then the initial phase (PND 25-26) started with 20 rats, and then 10 were placed in each control or experimental group (5 at PND 46, and 5 at PND72?). Also indicate the exact starting number of animals in the study, and if you had any losses (for example, died during the surgery/post-surgical period, failure in the AAV procedure, etc.). Please explain how the sample size was decided, and provide details of any a priori sample size calculation, if performed. Also, state if the housing of the animals had similar conditions (for example, all cages were in the same row at the animal facility, or some were up and others down), and if the cages’ location was randomized. Please state if the person in charge of water/ethanol administration to the rats was blinded or not.

o   Data acquisition: Please add the numerical aperture (NA) number of the microscope objectives used. In lines 246-247, the authors mention an algorithm for blind deconvolution, you could be a little bit more specific about the algorithm (or command) used? In lines 259-261, please specify how the fluorescent intensity was measured (same comment for the Neuroligins and Neurexin section). For sections 3.1 and 3.2 of the manuscript, please mention the sample size and number of images captured per brain area (in ACC, mPFC (infralimbic and prelimbic), OFC and LO-OFC), and the method used to capture them (i.e. images were randomly determined, automated or selected by the researcher).

o   Statistical analysis: Please include the normality tests used to evaluate if data behave or not as parametric. Please include how outliers were defined (and in the text mention if any value was removed as an outlier or not, and the rationale behind outlier remotion).

-          Please include the RRID numbers for antibodies and plasmids.

-          In line 185 the u should be substituted for the corresponding Greek letter. Also, correct this in the Figures and Figure legends of the paper.

-          In line 216, the degrees symbol is missing. Same issue in line 229.

-          The order or denomination of the figures is wrong as Figures 5 and 6 are in pages 8 and 9, respectively, and figures 2, 3 and 4 are in pages 11, 12 and 14, respectively.

-          For all the box plots, please include the individual data points in the figures.

-          Please add the mean ± SEM values in the text of the manuscripts for all experiments. Also, please add the statistical results for each comparison in the text (i.e. the exact p value), regardless of statistical significance.

-          In the figures, the 2 of the water formulae should be in subscript. Also, check this in the caption of the figures, including missing superscript as the one in figure 3.

-          The n number of all groups should be included in the captions of the figures.

-          Results from lines 285 to 286, 295 to 296, 402 to 404, and lines 440 to 442, could be annexed as supplementary data. In order to improve research reproducibility is recommended to avoid data exclusion, with statements such as “data not shown”. Many readers may find useful information in this data.

-          Please provide an explanation as why the dendritic spine analysis was only performed at PND 72 and not at PND 46.

-          I´m confused about the results from section 3.5. Blood EtOH Concentrations (BECs). I´m not sure if they were included in section 2.2. (lines 195-196), in which case they should be localized in the Results section of the paper, not in Methods, or if they were omitted from the paper, in which case they should be included in the Results section. Also, if results from lines 195-196 are not from this study, but from previous studies, please include the citations.

-          The citation in line 480 should be placed in line 478.

-          Please check the citation format in line 490.

-          The citation in line 531 should be placed in line 530.

-          In the last paragraph of the discussion, the authors may propose other alternatives to neurexin-neuroligin interactions (for example LRRTM2), as future targets to help explain some of the changes observed due to EtOH exposure. Similarly, authors may briefly mention if other EtOH-associated effects, such as inflammation may be responsible for some of the changes observed in the study.  

-          In the dendritic spine maturation experiments, at PND 72, less immature and more intermediate dendrites were observed in mPFC, and less intermediate and more mature dendrites were observed in the LO-OFC area. The meaning of these results should be included in the Discussion section.

-          I recommend the authors to include (either in the introduction or the discussion) the following recent papers about alcohol use/binge and astrocytes. I believe these papers can help to complement and improve the manuscript. The suggested papers are: (PMID: 35182671); (PMID: 33844860); (PMID: 30625475); (PMID: 29753887); (PMID: 29396480).

Author Response

We would like to thank Reviewer 1 for the insightful and thorough nature of this review, we appreciate the significant effort that was taken during this process. We have addressed all Reviewer 1 comments below. All responses are indicated in blue font in the text below and within the manuscript.

Comments:

  1. In lines 35-36, the paper by Patrick and Schulenberg is specific to adolescents in the USA. Please add this information to the phrase.

    The phrase, ‘in the United States’ has been added to the sentence.

  2. In lines 37-39, I suggest replacing the reference with the original paper (Pubmed number PMID: 31912524), as the included reference is focused on alcohol-related deaths during Covid-19 pandemic (although it is correctly used to reference the phrase at lines 41-43).

    Thank you for the suggestion, we have replaced the White et al., 2022 paper with the original White et al., 2020 paper.

  3. In lines 39-41, the WHO citation is missing from the References section of the paper. Please include this reference and double check if any other is missing. In line 54, Koob & Volkow citation is missing from the references section.

    WHO citation has been added to the reference section. All additional references have been checked to ensure that they are appropriately included in the reference section.

  4. In line 44, the word adolescents should be replaced with adolescence. Line 64, the dot should be substituted by a comma.

    These typos have been corrected. 
  5. Lines 56-57. Please correct this phrase.

    The grammatical error has been corrected.

  6. Lines 68-71. Please check the references. The terms infralimbic or prelimbic are not mentioned in the paper of Warren et al. (2016). It should be substituted for a more adequate reference.

The sentence has been rewritten and appropriate references have been added.

  1. Lines 108-110. Please check the numbers of single astrocyte interactions. The paper by Bushong et. Al (2002), mentions “an astrocyte occupying 66,000 μm3 of neuropil would oversee ∼140,000 synapses”, the number up to 100,000 is not mentioned in that paper. In addition, the paper of Bushong doesn´t mention 2 million synapses in humans, as this information was reported subsequently in the paper of Oberheim et al (2009) (PMID: 19279265), which should be cited in this phrase as well.

The statement referencing ‘100,000 synapses’ has been removed from the manuscript along with the Bushong et al. citation. The overall statement has been corrected and Oberheim et al (2009) has been added to the in-text citations and the reference page.

  1. Following the recommendations by ARRIVE 2.0 guidelines, SyRCLE and similar initiatives, I kindly suggest the authors to provide some additional information about the experimental procedures.

These resources are appreciated and additional experimental details have been included as recommended. Additional information includes details regarding housing, habituation, animal randomization, and blinding.

  1. Please add a clearer and more explicit description of the control and experimental groups used, indicating the number of animals per phase of the experimental procedure. For example, if you planned 5 animals per group and had 4 groups, then the initial phase (PND 25-26) started with 20 rats, and then 10 were placed in each control or experimental group (5 at PND 46, and 5 at PND72?).

The requested changes have been made and include the addition of total number of animals and animal number per group (see methods section). Animal numbers per group have also been added to the figure legends. A statement indicating that no animals were lost during the surgery or during post-surgical recovery has been added.

  1. Please explain how the sample size was decided, and provide details of any a priori sample size calculation, if performed.

A detailed description of the a priori sample size estimation has been included.

  1. Data acquisition: Please add the numerical aperture (NA) number of the microscope objectives used.

The NA has been added to the methods section.

  1. In lines 246-247, the authors mention an algorithm for blind deconvolution, you could be a little bit more specific about the algorithm (or command) used?

 We have added additional information regarding the algorithm, however it is limited due to the proprietary nature of the code (we do not own the rights to this software).

  1. In lines 259-261, please specify how the fluorescent intensity was measured (same comment for the Neuroligins and Neurexin section).

 Further details have been added to ensure replication is possible.

  1. For sections 3.1 and 3.2 of the manuscript, please mention the sample size and number of images captured per brain area (in ACC, mPFC (infralimbic and prelimbic), OFC and LO-OFC), and the method used to capture them (i.e. images were randomly determined, automated or selected by the researcher).

Apologies for the oversight, details regarding sample size and number of images captured per brain subregion have been added.

  1. Statistical analysis: Please include the normality tests used to evaluate if data behave or not as parametric. Please include how outliers were defined (and in the text mention if any value was removed as an outlier or not, and the rationale behind outlier remotion). Normality tests used during analysis have been added to the statistics section.

We use very stringent parameters for outlier removal and as a consequence, no data was removed.

  1. Please include the RRID numbers for antibodies and plasmids.

 RRIDs for antibodies and plasmids have been added.

  1. In line 185 the u should be substituted for the corresponding Greek letter. Also, correct this in the Figures and Figure legends of the paper.

u has been replaced by the corresponding Greek letter in all figures and figure legends.

  1. In line 216, the degrees symbol is missing. Same issue in line 229.

 This has been corrected.

  1. The order or denomination of the figures is wrong as Figures 5 and 6 are in pages 8 and 9, respectively, and figures 2, 3 and 4 are in pages 11, 12 and 14, respectively.

 The figures have been reorganized and replaced due to new analyses requested by Reviewer 2.

  1. For all the box plots, please include the individual data points in the figures.

 Individual data points have been included in the box plots.

  1. Please add the mean ± SEM values in the text of the manuscripts for all experiments. Also, please add the statistical results for each comparison in the text (i.e. the exact p value), regardless of statistical significance.

We began adding all mean ± SEM numbers to the body of the manuscript however, due to the new analysis request by Reviewer 2 it would now require the inclusion of an immense amount of data that significantly distracts from the significance, importance, and flow of the manuscript narrative. However, as a compromise ALL statistical comparisons are shown in the supplemental (Figure S1) along with all Mean ± SEMs (Figure S2), regardless of significance.

  1. In the figures, the 2 of the water formulae should be in subscript. Also, check this in the caption of the figures, including missing superscript as the one in figure 3.

Subscripts have been corrected.

  1. Results from lines 285 to 286, 295 to 296, 402 to 404, and lines 440 to 442, could be annexed as supplementary data. In order to improve research reproducibility it is recommended to avoid data exclusion, with statements such as “data not shown”. Many readers may find useful information in this data.

All non-significant data has been added to the supplemental in graph format, in addition to the inclusion of all statistical comparisons, and mean ± SEM.

  1. Please provide an explanation as why the dendritic spine analysis was only performed at PND 72 and not at PND 46.

 This has been addressed in the results section.

  1. I´m confused about the results from section 3.5. Blood EtOH Concentrations (BECs). I´m not sure if they were included in section 2.2. (lines 195-196), in which case they should be localized in the Results section of the paper, not in Methods, or if they were omitted from the paper, in which case they should be included in the Results section. Also, if results from lines 195-196 are not from this study, but from previous studies, please include the citations.

Apologies for this oversight, the BECs have been included in the results (section 5.1).

  1. Citation correction….line 480…line 490….line 531. should be placed in line 478.

All citations have been corrected and moved to the correct locations.

  1. In the last paragraph of the discussion, the authors may propose other alternatives to neurexin-neuroligin interactions (for example LRRTM2), as future targets to help explain some of the changes observed due to EtOH exposure. Similarly, authors may briefly mention if other EtOH-associated effects, such as inflammation may be responsible for some of the changes observed in the study.

LRRTM2 and neuroinflammation have been added to the discussion.

  1. In the dendritic spine maturation experiments, at PND 72, less immature and more intermediate dendrites were observed in mPFC, and less intermediate and more mature dendrites were observed in the LO-OFC area. The meaning of these results should be included in the Discussion section.

Potential reasons for these changes have been addressed in the discussion.

  1. I recommend the authors to include (either in the introduction or the discussion) the following recent papers about alcohol use/binge and astrocytes. I believe these papers can help to complement and improve the manuscript. The suggested papers are: (PMID: 35182671); (PMID: 33844860); (PMID: 30625475); (PMID: 29753887); (PMID: 29396480).

The suggested references have been added to the discussion.

Reviewer 2 Report

Overall, this is an interesting and timely study with its focus on changes in astrocyte structure following alcohol exposure. 

1. The use of gavage is obviously stressful and not a particularly valid model of ethanol consumption. Do the authors have data from naive animals to compare to the water controls to assure that any changes are due to ethanol and not the gavage protocol?

2. Data were all analyzed with t-tests when it appears that using a two-way Anova for each timepoint analyzed would be more appropriate. This would also determine if there were significant changes in astrocyte morphology/spine density across the various prefrontal regions. There are differences in the y-axis scales for some of these regions suggesting that there may be differences that were not  reported.

3. Please indicate in the analysis and figure legends the sample sizes and how many individual astrocytes per animal were used to calculate the data.

4. It appears that the Lack-GFP signal was obtained from the native fluorescence of the GFP. Do the authors have data to show that the fixation and antigen retrieval process did not alter this signal and that similar results would be obtained if a GFP antibody was used to amplify the signal?

Author Response

We thank Reviewer 2 for the thoughtful comments, particularly the suggestion to re-analyze the data in order to assess baseline-subregion differences. We have addressed all Reviewer 2 comments below. All responses are indicated in blue font in the text below and within the manuscript.

  1. The use of gavage is obviously stressful and not a particularly valid model of ethanol consumption. Do the authors have data from naive animals to compare to the water controls to assure that any changes are due to ethanol and not the gavage protocol?

    The Reviewer makes a valid point about the stressfulness of gavage. We have had concerns about this in the past. In order to address this, we always conduct water gavage for the control animals. We have previously conducted experiments with the addition of non-gavage controls. Although unpublished, we did not find any significant differences between water gavage and non-gavage rats in the final outcome measures. However, it should be noted that ethanol gavage is a very standard approach in the adolescent alcohol field. This approach is necessary because rats do not like to drink high quantities of alcohol and typically do not reach binge-level intoxication. A typical approach that results in higher drinking levels, is the ‘sucrose fade’ approach in which % sucrose is high and % alcohol is low at the beginning of voluntary drinking. Overtime, % sucrose is gradually reduced while % alcohol is gradually increased. However, this approach takes time and animals typically grow out of adolescents before rats can attain high enough alcohol consumption. It should also be noted that all other administration techniques bypass first-pass metabolism, a critical step in the processing of alcohol. Bypassing first-pass metabolism can significantly impact the rate of intoxication and metabolism (PMID:12798972).
  2. Data were all analyzed with t-tests when it appears that using a two-way Anova for each timepoint analyzed would be more appropriate. This would also determine if there were significant changes in astrocyte morphology/spine density across the various prefrontal regions. There are differences in the y-axis scales for some of these regions suggesting that there may be differences that were not reported.

 We thank the Reviewer for the suggestion. The data has been reanalyzed, using two-way ANOVA followed by Tukey’s post hoc analysis, revealing many interesting subregion differences throughout the manuscript that are now discussed.

  1. Please indicate in the analysis and figure legends the sample sizes and how many individual astrocytes per animal were used to calculate the data.

We apologize for this oversight. Sample sizes have now been added to the text and figure legends. We have also included all statistical analyses in the supplemental section.

  1. It appears that the Lck-GFP signal was obtained from the native fluorescence of the GFP. Do the authors have data to show that the fixation and antigen retrieval process did not alter this signal and that similar results would be obtained if a GFP antibody was used to amplify the signal?

We appreciate the intent of this comment, however Lck-GFP is inserted into an adenoassociated virus (AAV5) which is then intracranially injected, this is not natively expressed GFP fluorescence. We have to wait a minimum of 2 weeks for Lck-GFP to express within the membrane of the astrocyte. Since this is an AAV-mediated insertion resulting in strong fluorescence we have no need to amplify the signal with a GFP antibody. It should be noted, that it is necessary to use the AAV approach because astrocyte antibodies typically only target the backbone of the cell and are unable to capture the intricate nature of the finer astrocyte processes.

Round 2

Reviewer 1 Report

I appreciate the authors efforts to improve the manuscript. Certainly, this version is much clear and more complete. The paper improved a lot from the first document. I just noticed two very small details, in line 460 a greek letter is missing, and in line 540 a bracket is missing.

Reviewer 2 Report

The authors have been responsive to the previous critique and have made significant changes to the manuscript including a more appropriate analysis of the data. These changes improve the manuscript and there are only a few minor issues remaining.

1. Line 88-89: The OFC, which is critical for impulse control, has shown to be impaired in individuals with alcohol use disorders (McGuier et 89 al., 2015).... The McGuier reference cited here is incorrect as this study examined changes in dendritic spines in mice following chronic exposure to ethanol. A more appropriate reference should be cited to support this statement.

2. In the initial review, Reviewer 2 asked whether the fixation process could have affected the intrinsic Lck-GFP signal used to quantitate astrocyte volume. Based on their response, the authors likely misinterpreted this question. Although no changes are required, the issue is whether the results would have been similar if an antibody to GFP were used to detect the signal rather than the intrinsic fluorescence. While GFP is fairly resistant to fixation protocols, it can lose signal that can be recovered using an antibody.  
